# Quality of life and health status in older adults (≥65 years) up to five years following colorectal cancer treatment: Findings from the ColoREctal Wellbeing (CREW) cohort study

Amanda Cummings[1], Rebecca Foster[1], Lynn Calman[1], Natalia V. Permyakova[2,3], Jackie Bridges[4], Theresa Wiseman[5], Teresa Corbett[6], Peter W. F. Smith[7], Claire Foster[1] *

1 Macmillan Survivorship Research Group, School of Health Sciences, University of Southampton, Southampton, United Kingdom, 2 Southampton Clinical Trials Unit, Faculty of Medicine, University of Southampton, Southampton, United Kingdom, 3 NIHR Research Design Service South Central, Faculty of Medicine, University of Southampton, Southampton, United Kingdom, 4 NIHR ARC Wessex, School of Health Sciences, University of Southampton, Southampton, United Kingdom, 5 The Royal Marsden NHS Foundation Trust and University of Southampton, Southampton, United Kingdom, 6 Faculty of Sport, Health, & Social Sciences, Solent University, Southampton, United Kingdom, 7 Social Statistics and Demography, Social Sciences, University of Southampton, Southampton, United Kingdom

* C.L.Foster@soton.ac.uk

## Abstract

### Objective

Colorectal cancer (CRC) is common in older adults, with more than 70% of diagnoses in people aged ≥65 years. Despite this, there is a knowledge gap regarding longer-term outcomes in this population. Here, we identify those older people most at risk of poor quality of life (QoL) and health status in the five years following CRC treatment.

### Materials and methods

CREW is a UK longitudinal cohort study investigating factors associated with health and wellbeing recovery following curative-intent CRC surgery. Participants completed self-report questionnaires pre-surgery, then at least annually up to five years. Longitudinal analyses explored the prevalence and pre-surgery risk factors of poor QoL (QLACS-GSS) and health status (EQ-5D: presence/absence of problems in five domains) in older (≥65 years) participants over five years.

### Results

501 participants aged ≥65years completed questionnaires pre-surgery; 45% completed questionnaires five years later. Oldest-old participants (≥80 years) reported poorer QoL (18% higher QLACS-GSS) and 2–4 times higher odds of having problems with mobility or usual activities, compared with the youngest-old (65–69 years) over follow-up. Baseline higher self-efficacy was significantly associated with better QoL (10–30% lower QLACS-GSS scores compared to those with low self-efficacy) and lower odds of problems in all EQ-5D domains. Adequate social support was significantly associated with better QoL (8%

**Data Availability Statement:** Data cannot be shared publicly because of ethical restrictions. The data include sensitive patient information, including clinical, cancer related data. The ethical restrictions were imposed by the UK NHS Health Research Authority NRES Committee South Central - Oxford B (REC ref: 10/H0605/31)]. Data are available from the CentRIC+ Research Group Data Sharing Panel (CentRIC@soton.ac.uk) for researchers who meet the criteria for access to confidential data. The data underlying the results presented in the study are also available from the study webpage: http://www.horizons-hub.org.uk/access_data.html.

**Funding:** The ColoREctal Well-being (CREW) study is funded by Macmillan Cancer Support grant number 3546834. The funders had no role in study design, data collection and analysis, decision to publish, or preparation of the manuscript.

**Competing interests:** I have read the journal's policy and the authors of this manuscript have the following competing interests: Dr Lynn Calman has received an honorarium for teaching from Boehringer Ingelheim.

lower QLACS-GSS) and lower odds of problems with usual activities (OR = 0.62) and anxiety/depression (OR = 0.56).

## Conclusion

There are important differences in QoL and health status outcomes for the oldest-old during CRC recovery. CREW reveals pre-surgery risk factors that are amenable to intervention including self-efficacy and social support.

## Introduction

Colorectal cancer (CRC) is the third most common cancer worldwide and one of the most commonly diagnosed malignancies in the older population [1, 2]. In the UK, more than 70% of CRC diagnoses occur in people aged ≥65 [3]. With increasing survival rates (57% surviving ≥10 years) in an ageing population, almost two thirds of people living with and beyond (referred to herein as 'living with') cancer are aged over 65 years [4, 5].

Current evidence shows that older adults after cancer diagnosis report poorer QoL and health status compared to older adults without a cancer diagnosis [6–10]. Similarly, in breast cancer, newly diagnosed older women report poorer functioning than younger counterparts [6]. In order to improve outcomes, research into older adults living with CRC is needed to identify individuals at greatest risk of poorer health and wellbeing over time [7]. However, despite being the fastest growing age group living with cancer, the predominance of older people is often not reflected in current research [11]. This under-representation has limited our understanding of their distinct needs, resulting in deficiencies in tailored intervention development and support for this growing population [9, 12]. As a result, older people, often with additional complications caused by altered physiology, multimorbidity, functional and cognitive impairments, are often inadequately supported by healthcare [9, 13]. It has been suggested that the long-term effects of cancer alongside the physical and psychosocial changes associated with ageing, may exacerbate problems for older people [11]. However, there is limited evidence regarding their longer-term QoL and functioning outcomes [14–16]; including who is at greatest risk of poor health and wellbeing outcomes [9, 14]. Understanding the specific issues faced by older adults and improving their quality of life (QoL), are globally recognised research priorities [12, 17].

We address the knowledge gap and research priorities outlined above, using data from the ColoREctal Wellbeing study (CREW); a prospective longitudinal cohort study investigating factors associated with recovery of health and wellbeing in the five years following CRC [18]. This study aims to investigate the differences in pre-surgical (sociodemographic, clinical, treatment, environmental and personal) factors across older people (≥65 years) diagnosed with CRC and their associations with post-surgical QoL and health status up to five years following treatment for CRC.

## Materials and methods

### Study design, participants and procedure

CREW is a multi-centre, longitudinal cohort study of patients with non-metastatic CRC. Participants were recruited between 2010 and 2012 from 29 cancer centres across the UK. Eligible

participants were those aged ≥18 years (no upper limit) being treated with curative-intent surgery for CRC (Dukes' stage A-C), and able to complete questionnaires. Distant metastatic disease at diagnosis or a prior cancer diagnosis were exclusion criteria. All eligible patients were invited to participate. Participants completed self-report questionnaires at baseline (pre-surgery), 3, 9, 15, 24 months, and annually up to 5 years. Clinical characteristics were collected from NHS medical data. Additional details are described elsewhere [18]. Ethical approval was granted by the UK NHS Health Research Authority NRES Committee South Central—Oxford B (REC ref: 10/H0605/31).

Whilst the definition of 'older adult' lacks uniform consensus, ≥65 years is a commonly adopted threshold in cancer literature [6, 11]. We therefore report data pertaining to CREW participants aged ≥65 on the date of baseline questionnaire completion in our description of 'older participants'.

## Measures

Foster and Fenlon's conceptual model of health and wellbeing recovery following cancer treatment informed data collection in CREW [19]. Their model recognises the importance of multiple factors in contributing toward the recovery of health and wellbeing, including socio-demographic (referred to as 'pre-existing' in the model), clinical, treatment, environmental and personal factors (Fig 1). For this publication, five domains of assessment were selected, each with corresponding predictors. Fig 1 shows these variables mapped onto the conceptual model. Each predictor holds significance as an identifiable, or potentially modifiable factor, for informing and targeting future interventions for the health and well-being of older people with CRC.

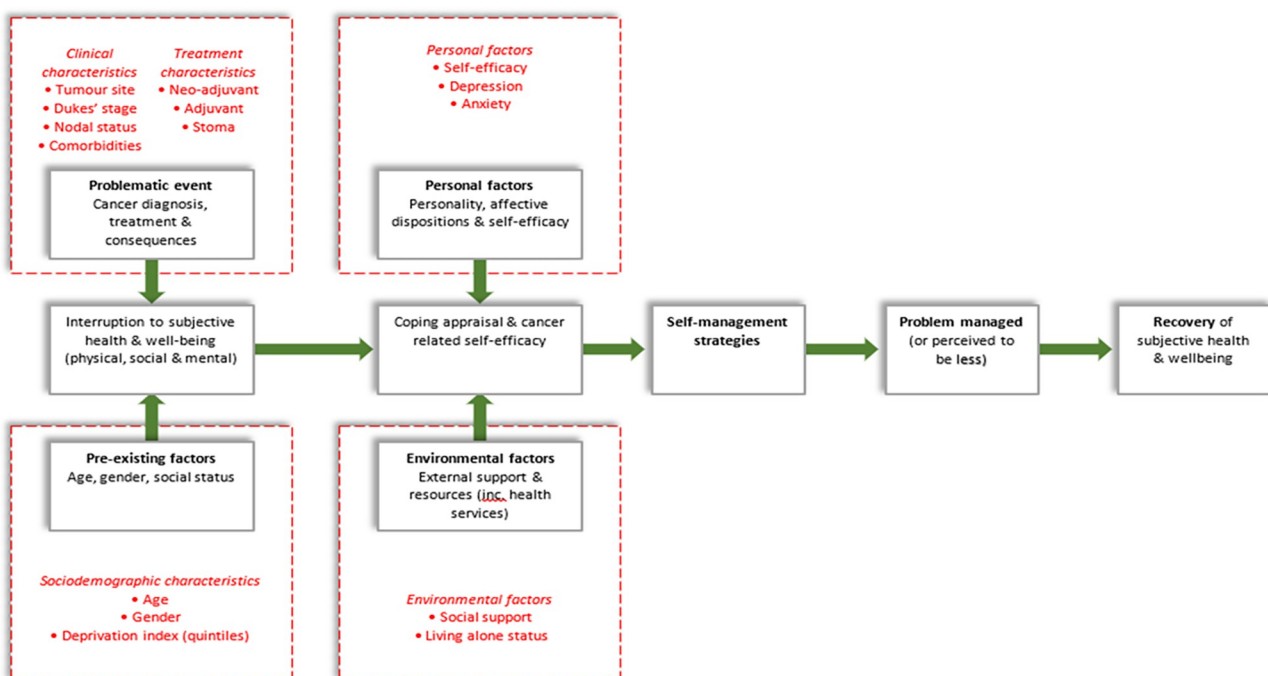

**Fig 1. Predictors for older people with CRC mapped onto Foster and Fenlon's conceptual model of health and wellbeing recovery following cancer treatment.**

Full details of all CREW measures are published elsewhere [18]. Information below pertains to measures presented in this paper.

## Outcome measures

Areas of assessment reflect their relevance to an older cancer population. The Quality of Life in Adult Cancer Survivors (QLACS) measure assesses to what degree individuals are *bothered* by a problem [20]. The EQ-5D captures impaired health status and the subsequent *impact* on an individual's everyday life [21]. Both measures were collected at baseline and every subsequent timepoint (apart from EQ-5D at 9 months).

**Quality of life.** QLACS assesses health-related quality of life among long-term cancer survivors [20]. QLACS Part 1 measures generic quality of life with 28 items organised into 8 domains using a 7 point rating scale [20]. The sum of these domains yields a Generic Summary Score (GSS). Informed by the underpinning conceptual model above, QLACS-GSS was the primary outcome measure in CREW [18].

**Health status.** EQ-5D provides a generic assessment of self-rated health status, and is widely adopted in cancer survivor populations [21]. The measure consists of five domains (mobility, self-care, usual activities, pain/discomfort, anxiety/depression) each scored on a 3-point scale as none/some/severe problems. Due to low counts of participants with 'severe problems', each EQ-5D domain was dichotomised as presence/absence of problems.

## Predictors

**Sociodemographic characteristics.** Gender was self-reported by participants in questionnaires. Older participants were divided into four sub-groups by age (65–69, 70–74, 75–79, ≥80 years). Neighbourhood deprivation was derived from postcodes using the index of multiple deprivation and converted into quintiles [22].

**Clinical and treatment characteristics.** Clinical and treatment data were acquired from medical notes (with consent): tumour site (colon, rectum), Dukes' stage (A, B, C), nodal involvement (N0, N1-2), presence of a stoma (yes, no) and (neo)adjuvant treatment details (none, any (chemotherapy, radiotherapy or both)). Participants self-reported the presence of comorbidities in questionnaires. The study-specific comorbidity measure related to twelve individual physical and mental health conditions or disease groups and asked whether a doctor had ever told the participant they had the condition and whether the condition limited their typical daily activities [23].

**Environmental factors.** Social support was assessed using the Medical Outcomes Study (MOS) 19-item scale, where higher scores indicate greater social support (range 0–100) [24]. An overall score of ≥ 80 was considered good social support [25]. Whether or not participants lived alone was assessed by self-report in questionnaires.

**Personal factors.** Personal factors were assessed using validated measures in the self-report questionnaires. Depression and anxiety were assessed using the Centre for Epidemiological Studies Depression Scale (CES-D) [26] and the State-Trait Anxiety Inventory (STAI) [27], respectively. Both are 20-item measures, where higher scores indicate greater depression or anxiety. Scores ≥40 on the STAI suggest clinically significant anxiety and scores of ≥20 on the CES-D suggest clinical depression. Self-efficacy was assessed using the Self-efficacy for Managing Chronic Disease (SEMCD) Scale, a 6-item measure using a rating scale from 1–10. The mean of all items provides an overall score, with higher scores indicating greater confidence to manage illness-related problems [28]. The following cut-offs were assigned to reflect different levels of self-efficacy: 1–4 'low confidence', 5–6 'moderate confidence', 7–8 'confident' and 9–10 'very confident' [29].

## Statistical analysis

All predictors and outcomes were assessed at baseline (pre-surgery), except for comorbidities and living alone status (questionnaire self-report at 3 months). Several variables were assessed at multiple timepoints post-surgery: QLACS-GSS, five EQ-5D domains, living alone status, MOS, CES-D, STAI and SEMCD. Continuous variables were categorised according to the guidelines on clinically significant cut offs (see Measures). There was less than 5% of missing data in each variable, hence, no categories for missing data were included for any variable (but accounted for in the column percentages in the descriptive output).

Descriptive analyses compared baseline values of all variables for four age sub-groups of older participants (65–69, 70–74, 75–79, ≥80 years). We also compared baseline values of the outcomes with their values at each follow-up timepoint. Using Shapiro–Wilk and Shapiro–Francia tests, preliminary normality assessment revealed a non-parametric distribution in QLACS-GSS by each age group. Therefore, a median with lower and upper quartiles was indicated for QLACS-GSS using a Kruskal-Wallis rank test in the four age sub-groups and Mann-Whitney test in the comparison with follow-up timepoints. The differences in categorical variables by each grouping type were assessed using a chi-squared test. In addition, similar descriptive analyses were conducted comparing older (≥65) and younger (<65) CREW participants to add context to our findings for older people.

Multivariable regression analyses were conducted for the QLACS-GSS and each EQ-5D domain to identify those baseline risk factors which were associated with poorer outcomes for older participants over the 5-year follow-up. A log-linear regression model was applied for QLACS-GSS to meet the assumption of normal distribution. Each EQ-5D domain had a logistic regression model fitted to predict the likelihood of presence of problems separately in each domain. STAI and CES-D were not included as predictors in the regression model of 'Anxiety/Depression' EQ-5D domain due to their high correlation with this outcome.

To be able to observe the associations between pre-surgery predictors and post-surgery outcomes, individuals who participated at baseline and at least one other timepoint were included in the regression analyses. Descriptive statistics accounted for anyone who completed the baseline questionnaire and reported their date of birth. There was no imputation of missing data, however, to minimise the attrition bias, a population-average approach was applied in each regression model to account for multiple observations of the same participants over 5-year follow-up post-surgery; therefore, the time of participation after baseline was controlled for in each model. A backward stepwise selection criterion was applied in the regression analyses of each outcome allowing only statistically significant (p<0.05) confounders to be carried forward in two steps: first, it was applied in each regression model per domain (see S1 Appendix); second, the statistically significant predictors across all domains from step one were carried forward and tested for significance in a single regression model per outcome. All analyses were carried out at the 5% significance level using Stata Corp. StataSE 14.

## Results

### Participants

The CREW cohort is a representative sample of eligible patients treated during the recruitment period. Recruitment and participation rates are described in detail elsewhere [30]. Fig 2 shows the numbers of patients who were screened, eligible and who gave consent to take part in the study. Of those eligible (n = 1,350), 78% agreed to participate, 909 gave consent to receive questionnaires and 146 gave consent for the collection of clinical data only. Excluding 48 individuals who were found to be ineligible following surgery, or who withdrew or died between

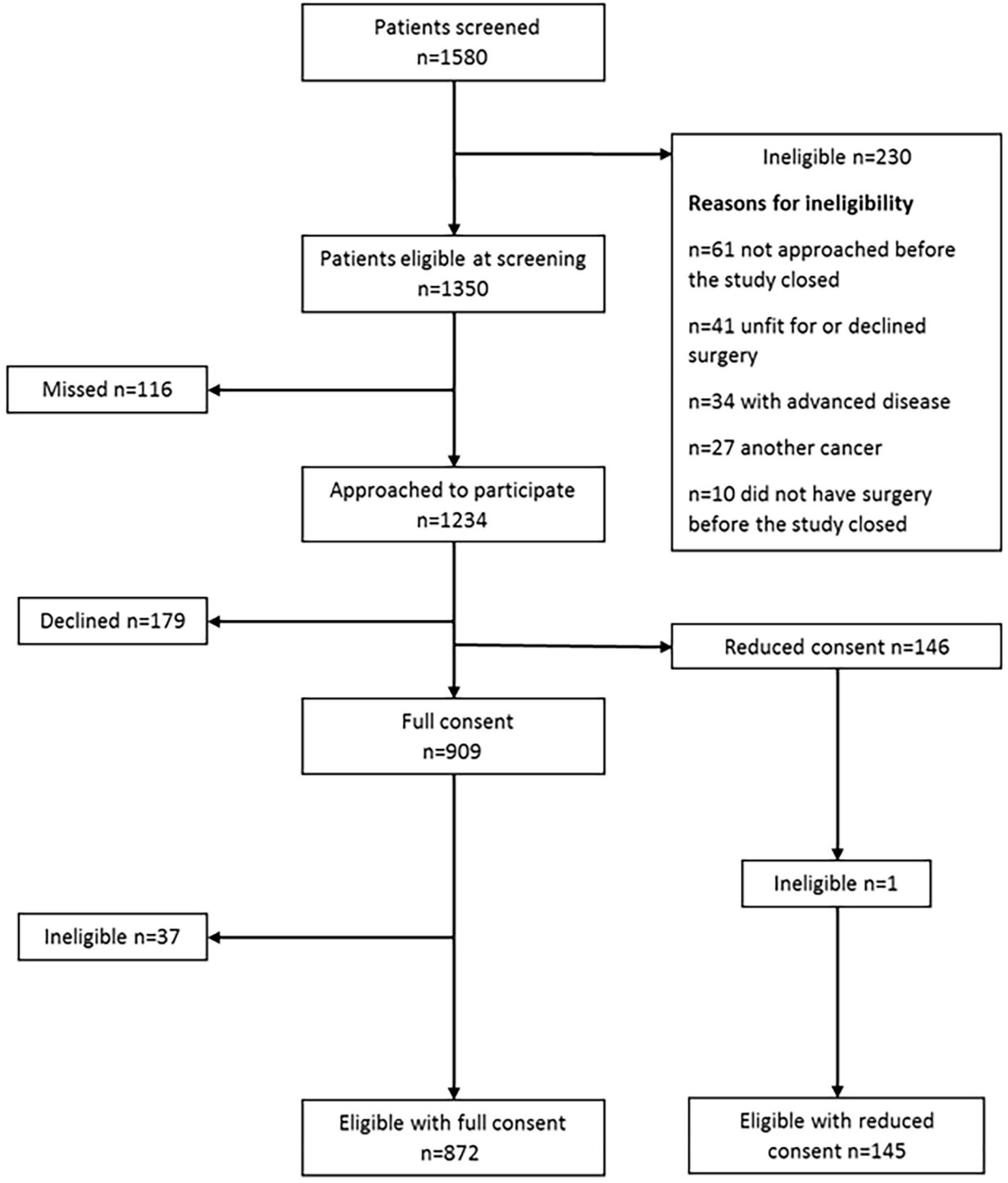

**Fig 2. Flowchart showing screening of patients, eligibility and numbers consenting.**

consent and baseline, a total of 861 eligible participants consented to questionnaire follow-up. The baseline return rate was 88% (756/861) and 5-year return rate 43% (371/861). The mean age of participants at baseline was 68 (SD = 10.3, range 32–95) and 68% (584/861) were aged ≥65. Of these older participants aged ≥65, 133 (23%) died, 135 (23%) actively withdrew and

90 (15%) were lost to follow-up during the five-year period. Of the 584 participants aged ≥65, 86% returned a questionnaire (501/584) at baseline and 39% (226/584) at five years. Four hundred and fifty-one older participants returned questionnaires at baseline and at least one other timepoint.

## Baseline characteristics

Of participants aged ≥65 at baseline (n = 501), 26% were aged 70–74, 19% aged 75–79 and 19% aged ≥80. Sixty percent were male, 20% were living alone and 22% reported at least one comorbidity which the participant felt 'limited their typical daily activities'. Most participants had colon cancer (67%), Dukes' stage B (56%) and absence of nodal involvement (65%) (Table 1).

Comparisons between characteristics of older (≥65years) and younger (<65years) CREW participants pre-surgery are shown in S2 Appendix. In summary, older participants reported better QoL than younger participants but were more likely to experience problems with mobility and self-care.

**Comparisons by older age sub-groups (65–69, 70–74, 75–79 and ≥80 years) at baseline.** The oldest-old, aged ≥80, were found to differ significantly from the those aged 65 to 69 in several respects at baseline (pre-surgery). Among participants aged ≥80, twice as many reported mobility problems (EQ-5D mobility), compared with those aged 65 to 69 (Table 1, p = 0.001). The ≥80 age group were also significantly more likely to live alone (p<0.001), not have adequate social support (p = 0.001) and not report high self-efficacy (confidence to manage illness-related problems, p = 0.038) at baseline. In addition, clinical and treatment data analysis revealed that receipt of any adjuvant therapy was almost five times lower among the oldest-old in comparison to the youngest-old (p<0.001).

## Quality of life (QLACS-GSS) over time

QoL in those aged ≥65 worsened over the 9 months following surgery and then improved to better than pre-treatment levels at 15 months, before levelling off (S3 Appendix). The best QoL for older people was experienced between 15 and 24 months.

We identified, using a log-linear regression model, several pre-surgery characteristics of participants aged ≥65 which predicted QoL scores in the 5 years following surgery. As shown in Table 2, predictors of lower QoL, included: being aged ≥80, having at least one comorbidity which the participant felt 'limited their typical daily activities', not having adequate social support, having low self-efficacy (confidence to manage illness-related problems) and having a clinically significant level of anxiety or depression. Further details of the associations which emerged are as follows. The oldest-old (aged ≥80) reported on average 18% (95%CI: 9.05 to 28.58) worse QLACS-GSS scores compared to youngest-old (aged 65 to 69). Participants with at least one limiting comorbidity, reported on average 16% (95%CI: 5.97 to 26.47) worse QoL scores. Adequate social support pre-surgery was significantly associated with better QoL (8% better score) over follow-up (95%CI: 13.59 to -2.24). Pre-surgery self-efficacy showed the greatest association with QoL over follow-up. Compared to those with low confidence, each group of participants with higher levels of self-efficacy (moderately confident, confident, and very confident) reported significantly better QoL over five years post-surgery: on average 10% (95%CI: 18.77 to -0.58), 16% (95%CI: 24.26 to -7.65) and 30% (95%CI: 37.51 to -22.52) lower QoL scores, respectively. Clinically significant levels of both depression and anxiety pre-surgery were significantly associated with poorer QoL over time: on average 20% (95%CI: 9.97 to 30.42) and 11% (95%CI: 2.76 to 19.82) lower QoL scores, respectively.

**Table 1. Distribution of baseline characteristics for total older CREW participants (≥65 years) and by four age sub-groups for older participants.**

| Baseline Characteristics | | Total Older ≥65 years) Participants (n = 501, 100%) | Aged 65–69 (n = 178, 35.53%) | Aged 70–74 (n = 132, 26.35%) | Aged 75–79 (n = 97, 19.36%) | Aged ≥80 (n = 94, 18.76%) | P-value[1] |
|---|---|---|---|---|---|---|---|
| | | *median (LQ, UQ) / n (%)* | *median (LQ, UQ) / n (%)* | *median (LQ, UQ) / n (%)* | *median (LQ, UQ) / n (%)* | *median (LQ, UQ) / n (%)* | |
| Outcomes | *QLACS-GSS* | *67.0 (52.0, 82.0) [n = 436]* | *64.33 (49.00, 81.00) [n = 167]* | *67.50 (52.00, 82.00) [n = 114]* | *66.33 (54.50, 81.50) [n = 84]* | *71.00 (60.00, 84.00) [n = 71]* | *0.172* |
| | **EQ-5D Mobility** | | | | | | |
| | Having no problems | 365 (72.85) | 143 (80.34) | 95 (71.97) | 72 (74.23) | 55 (58.51) | 0.001 ** |
| | Having problems | 129 (25.75) | 34 (19.10) | 32 (24.24) | 25 (25.77) | 38 (40.43) | |
| | **EQ-5D Self-care** | | | | | | |
| | Having no problems | 446 (89.02) | 156 (87.64) | 117 (88.64) | 93 (95.88) | 80 (85.11) | 0.050 |
| | Having problems | 47 (9.38) | 20 (11.24) | 10 (7.58) | 4 (4.12) | 13 (13.83) | |
| | **EQ-5D Usual activities** | | | | | | |
| | Having no problems | 323 (64.47) | 118 (66.29) | 84 (63.64) | 71 (73.20) | 50 (53.19) | 0.097 |
| | Having problems | 172 (34.33) | 59 (33.15) | 45 (34.09) | 25 (25.77) | 43 (45.74) | |
| | **EQ-5D Pain/ discomfort** | | | | | | |
| | Having no problems | 239 (47.70) | 95 (53.37) | 58 (43.94) | 47 (48.45) | 39 (41.49) | 0.225 |
| | Having problems | 253 (50.50) | 81 (45.51) | 69 (52.27) | 49 (50.52) | 54 (57.45) | |
| | **EQ-5D Anxiety/ depression** | | | | | | |
| | Having no problems | 348 (69.46) | 125 (70.22) | 89 (67.42) | 69 (71.13) | 65 (69.15) | 0.836 |
| | Having problems | 147 (29.43) | 51 (28.65) | 40 (30.30) | 27 (27.84) | 29 (30.85) | |
| Block 1—socio-demo | **Gender** | | | | | | |
| | Male | 301 (60.08) | 119 (66.85) | 69 (52.27) | 59 (60.82) | 54 (57.45) | 0.070 |
| | Female | 200 (39.92) | 59 (33.15) | 63 (47.73) | 38 (39.18) | 40 (42.55) | |
| | **Deprivation index** | | | | | | |
| | 1st quintile—least deprived | 95 (18.96) | 28 (15.73) | 26 (19.70) | 23 (23.71) | 18 (19.15) | 0.447 |
| | 2nd quintile | 106 (21.26) | 38 (21.35) | 29 (21.97) | 20 (20.62) | 19 (20.21) | |
| | 3rd quintile | 99 (19.76) | 33 (18.54) | 22 (16.67) | 23 (23.71) | 21 (22.34) | |
| | 4th quintile | 92 (18.36) | 31 (17.42) | 29 (21.97) | 13 (13.40) | 19 (20.21) | |
| | 5th quintile—most deprived | 101 (22.16) | 46 (25.84) | 23 (17.42) | 18 (18.56) | 14 (14.89) | |
| Block 2—environ-mental | **Living alone status** | | | | | | |
| | No | 290 (57.88) | 123 (69.10) | 77 (58.33) | 64 (65.98) | 38 (40.43) | <0.001 *** |
| | Yes | 100 (19.96) | 26 (14.61) | 29 (21.97) | 13 (13.40) | 32 (34.04) | |
| | Unknown/no-3m | 111 (22.16) | 29 (16.29) | 26* (19.70) | 20 (20.62) | 24 (25.53) | |
| | *Social support (MOS)* | | | | | | |
| | Inadequate (MOS<80) | 190 (37.92) | 49 (27.53) | 52 (39.39) | 38 (39.18) | 51 (54.26) | 0.001 ** |
| | Adequate (MOS ≥80) | 306 (61.08) | 127 (71.35) | 80 (60.61) | 58 (59.79) | 41 (43.62) | |

*(Continued)*

**Table 1.** (Continued)

| Baseline Characteristics | | Total Older ≥65 years) Participants (n = 501, 100%) | Aged 65–69 (n = 178, 35.53%) | Aged 70–74 (n = 132, 26.35%) | Aged 75–79 (n = 97, 19.36%) | Aged ≥80 (n = 94, 18.76%) | P-value[1] |
|---|---|---|---|---|---|---|---|
| | | *median (LQ, UQ) / n (%)* | *median (LQ, UQ) / n (%)* | *median (LQ, UQ) / n (%)* | *median (LQ, UQ) / n (%)* | *median (LQ, UQ) / n (%)* | |
| **Block 3—clinical** | **Tumour site** | | | | | | |
| | Colon | 338 (67.47) | 117 (65.73) | 90 (68.18) | 65 (67.01) | 66 (70.21) | 0.384 |
| | Rectum | 161 (32.14) | 61 (34.27) | 40 (30.30) | 32 (32.99) | 28 (29.79) | |
| | **Dukes' stage** | | | | | | |
| | stage A | 68 (13.57) | 19 (10.67) | 18 (13.64) | 17 (17.53) | 14 (14.89) | 0.721 |
| | stage B | 282 (59.29) | 104 (58.43) | 78 (59.09) | 51 (52.58) | 49 (52.13) | |
| | stage C | 142 (28.34) | 52 (29.21) | 32 (24.24) | 28 (28.87) | 30 (31.91) | |
| | **Nodal status** | | | | | | |
| | N0 | 327 (65.27) | 114 (64.04) | 92 (69.70) | 64 (65.98) | 57 (60.64) | 0.887 |
| | N1-N2 | 139 (27.74) | 52 (29.21) | 32 (24.24) | 26 (26.80) | 29 (30.85) | |
| | **Any comorbidities** | | | | | | |
| | None | 84 (16.77) | 34 (19.10) | 21 (15.91) | 12 (12.37) | 17 (18.09) | 0.420 |
| | Yes, non-limiting | 142 (28.34) | 74 (41.57) | 51 (38.64) | 34 (35.05) | 25 (26.60) | |
| | Yes, limiting | 111 (22.16) | 34 (19.10) | 30 (22.73) | 25 (25.77) | 22 (23.40) | |
| | Unknown/no-3m | 164 (32.73) | 29 (16.29) | 26 (19.70) | 20 (20.62) | 23 (24.47) | |
| **Block 4—treatment** | **Stoma** | | | | | | |
| | Yes | 157 (31.34) | 59 (33.15) | 38 (28.79) | 32 (32.99) | 28 (29.79) | 0.447 |
| | No | 337 (67.27) | 118 (66.29) | 90 (68.18) | 65 (67.01) | 64 (68.09) | |
| | **Neoadjuvant therapy** | | | | | | |
| | None | 413 (82.44) | 145 (81.46) | 108 (81.82) | 76 (78.35) | 84 (89.36) | 0.137 |
| | Any (CT, RT, both) | 84 (16.77) | 32 (17.98) | 21 (15.91) | 21 (21.65) | 10 (10.64) | |
| | **Adjuvant therapy** | | | | | | |
| | None | 353 (70.46) | 110 (61.80) | 86 (65.15) | 71 (73.20) | 86 (91.49) | <0.001 *** |
| | Any (CT, RT, both) | 146 (29.14) | 68 (38.20) | 44 (33.33) | 26 (26.80) | 8 (8.51) | |
| **Block 5—Personal factors** | **Self-efficacy (LORIG)** | | | | | | |
| | Low confidence | 56 (11.18) | 14 (7.87) | 15 (11.36) | 12 (12.37) | 15 (15.96) | 0.038 * |
| | Moderate confidence | 102 (20.36) | 30 (16.85) | 27 (20.45) | 20 (20.62) | 25 (26.60) | |
| | Confident | 214 (42.71) | 91 (51.12) | 49 (37.12) | 39 (40.21) | 35 (37.23) | |
| | Very confident | 118 (23.55) | 40 (22.47) | 39 (29.55) | 25 (25.77) | 14 (14.89) | |
| | **Clinical depression (CES-D)** | | | | | | |
| | No (<20 CES-D) | 394 (78.64) | 148 (83.15) | 102 (77.27) | 77 (79.38) | 67 (71.28) | 0.205 |
| | Yes (> = 20 CES-D) | 93 (18.56) | 27 (15.17) | 28 (21.21) | 16 (16.49) | 22 (23.40) | |
| | **Clinical anxiety (STAI)** | | | | | | |
| | No (<40 STAI) | 316 (63.07) | 119 (66.85) | 77 (58.33) | 62 (63.92) | 58 (61.70) | 0.663 |
| | Yes (> = 40 STAI) | 173 (34.53) | 56 (31.46) | 52 (39.39) | 33 (34.02) | 32 (34.04) | |

*Note*: missing values contributed less than 5% in each variable (not presented, but accounted for in the column percentages); LQ = lower quartile; UQ = upper quartile

[1] a Kruskal-Wallis rank test for continuous variables (all had a non-parametric distribution per age group confirmed by the Shapiro–Wilk and Shapiro–Francia tests) and a chi-squared test for categorical variables were used to test the equality-of-populations by four age groups of older people.

***p<0.001,

**p<0.01,

* p<0.05

**Table 2. Log-linear regression model predicting the QLACS-GSS score over the five-year follow-up post-surgery, predictors collected at baseline (pre-surgery), presented coefficients converted to percentage change and 95% confidence intervals (CI).**

| Predictors at Baseline | Percentage Change (95% CI) |
|---|---|
| Age (ref: 65–69 years) | |
| *70–74 years* | 1.17 (-5.80 to 8.65) |
| *75–79 years* | 2.43 (-5.54 to 11.07) |
| *≥80 years* | 18.41*** (9.05 to 28.58) |
| Comorbidities (ref: none) | |
| *≥1 non-limiting comorbidity* | -0.03 (-8.07 to 8.71) |
| *≥1 limiting comorbidity* | 15.77** (5.97 to 26.47) |
| *Not known* | 4.50 (-4.17 to 13.95) |
| Adequate social support (ref: MOS<80) | -8.09** (-13.59 to -2.24) |
| Self-efficacy (ref: low confidence) | |
| *Moderately confident* | -10.14* (-18.77 to -0.58) |
| *Confident* | -16.37*** (-24.26 to -7.65) |
| *Very confident* | -30.42*** (-37.51 to -22.52) |
| Clinically significant depression (ref: CES-D<20) | 19.76*** (9.97 to 30.42) |
| Clinically high anxiety (ref: STAI<40) | 10.96** (2.76 to 19.82) |

Note:

*$p < 0.05$,

**$p < 0.01$,

***$p < 0.001$.

Adjusted for the waves of participation after baseline (at least one wave between 3m and 60m).

## Health status (EQ-5D) outcomes over time

Problems with pain or discomfort and anxiety or depression significantly decreased over follow-up (S4 Appendix), with a decrease of around 40% at five years for both domains in comparison to pre-surgery levels. Problems in the other three EQ-5D domains (mobility, self-care and usual activities) increased significantly in the three months post-surgery then improved (self-care and usual activities) or remained higher than baseline (mobility).

Pre-surgery characteristics of participants aged ≥65 which predicted health related problems in the five years following surgery for CRC were identified using a logistic regression model (Table 3). Predictors included being older, living in a deprived geographical area, having at least one comorbidity which the participant felt 'limited their typical daily activities', receiving adjuvant chemo and/or radiotherapy, having a stoma, not having adequate social support and having low self-efficacy (confidence to manage illness-related problems). Further details of the associations which emerged are as follows. Being older predicted greater problems with mobility and usual activities over follow-up. This was most significant for oldest-old participants ($p < 0.001$) whose odds of reporting a problem with mobility or usual activities were four and two times higher respectively, in comparison with the youngest-old. Participants in the most deprived quintile had between two- and three-times higher odds of reporting problems with mobility, self-care and usual activities, in comparison with the least deprived quintile. Participants with at least one limiting comorbidity pre-surgery had five times higher odds of reporting at least some problem with mobility over follow-up, and 3–4 times higher odds of having problems with usual activities and pain/discomfort. Participants who received adjuvant therapy (chemo- and/or radiotherapy) had nearly twice higher odds of reporting problems with self-care over five years post-surgery. Participants without a stoma had

**Table 3. Logistic regression models predicting health problems separately in five EQ-5D domains over the five-year follow-up post-surgery, predictors collected at baseline (pre-surgery), presented coefficients converted to odds ratios (ORs) and 95% confidence intervals (CI).**

| Predictors at Baseline | Health Status (EQ-5D) Outcomes OR (95% CI) | | | | |
|---|---|---|---|---|---|
| | Mobility | Self-care | Usual activities | Pain/ discomfort | Anxiety/ depression |
| Age (ref: 65–69 years) | | | | | |
| *70–74 years* | 1.60 (0.93 to 2.76) | | 1.08 (0.68 to 1.69) | | |
| *75–79 years* | 1.92* (1.12 to 3.30) | | 1.71* (1.02 to 2.89) | | |
| *≥80 years* | 4.34*** (2.36 to 7.98) | | 2.49** (1.38 to 4.49) | | |
| Deprivation (ref: 1st quintile—least deprived) | | | | | |
| *2nd quintile* | 1.44 (0.76 to 2.75) | 1.34 (0.56 to 3.20) | 2.07* (1.11 to 3.83) | | |
| *3rd quintile* | 1.74 (0.93 to 3.24) | 1.81 (0.68 to 4.85) | 3.10*** (1.75 to 5.50) | | |
| *4th quintile* | 1.35 (0.68 to 2.66) | 2.09 (0.86 to 5.07) | 2.73** (1.49 to 5.01) | | |
| *5th quintile—most deprived* | 2.42** (1.28 to 4.59) | 2.72* (1.16 to 6.40) | 2.58** (1.42 to 4.69) | | |
| Comorbidities (ref: none) | | | | | |
| *≥1 non-limiting comorbidity* | 1.34 (0.71 to 2.52) | | 1.16 (0.66 to 2.05) | 1.10 (0.65 to 1.84) | |
| *≥1 limiting comorbidity* | 5.04*** (2.66 to 9.54) | | 3.92*** (2.20 to 7.01) | 3.45*** (2.02 to 5.88) | |
| *Not known* | 1.15 (0.58 to 2.28) | | 1.03 (0.56 to 1.90) | 1.21 (0.70 to 2.09) | |
| No stoma (ref: presence of stoma) | | | 0.50*** (0.34 to 0.74) | 0.64* (0.44 to 0.92) | |
| Adjuvant therapy (ref: none) | | 1.95* (1.08 to 3.54) | | | |
| Adequate social support (ref: MOS<80) | | | 0.62* (0.42 to 0.91) | | 0.56** (0.37 to 0.84) |
| LORIG (ref: low confidence) | | | | | |
| *Moderately confident* | 0.47* (0.26 to 0.97) | 0.64 (0.27 to 1.50) | 0.83 (0.43 to 1.62) | 0.52* (0.27 to 0.10) | 0.67 (0.35 to 1.27) |
| *Confident* | 0.30*** (0.16 to 0.58) | 0.30** (0.14 to 0.63) | 0.49* (0.26 to 0.93) | 0.38** (0.21 to 0.68) | 0.29*** (0.16 to 0.53) |
| *Very confident* | 0.18*** (0.09 to 0.39) | 0.11*** (0.05 to 0.26) | 0.22*** (0.11 to 0.44) | 0.28*** (0.15 to 0.54) | 0.18*** (0.09 to 0.36) |

*p<0.05,

**p<0.01,

***p<0.001.

Each model was adjusted for the waves of participation after baseline (at least one wave between 3m and 60m). The Anxiety/Depression outcome excluded CES-D and STAI from its list of predictors due to collinearity.

significantly lower odds of reporting problems with usual activities (OR = 0.50, 95%CI: 0.34 to 0.74) and pain/discomfort (OR = 0.64, 95%CI: 0.44 to 0.92). Adequate social support pre-surgery was predictive of less problems with usual activities (OR = 0.62, 95%CI: 0.42 to 0.91) and anxiety/depression (OR = 0.56 95%CI: 0.37 to 0.84). Self-efficacy demonstrated significant associations with each of the EQ-5D domains. Compared to those with low confidence pre-surgery, very confident participants had between 70% and 90% lower odds of reporting any post-surgery problems with mobility, self-care, usual activities, pain/discomfort and anxiety/depression over the five-year follow-up.

## Discussion

Our findings support the existing evidence that older people report better QoL compared with younger people following CRC [31–33]. However, our results also indicate that these broad comparisons mask important differences in QoL and health status within the ≥65s. The CREW study reveals that of those aged ≥65, the oldest individuals, who have typically been excluded from research studies [34], are the most vulnerable to poorer recovery outcomes. We reveal several areas that are amenable to intervention that if addressed appropriately could significantly improve recovery experiences for the oldest people living with and beyond CRC.

## Oldest old

Improved psychosocial adaptation and differences in the appraisal of stressful life events with age offer a possible explanation for individuals aged ≥65, as a whole, experiencing better QoL than those aged <65 years [31–34]. Existing literature highlights the health, functioning, and psychosocial differences between age groupings of older adults, but is limited by inclusion of only the youngest-old [6, 35]. The inclusion of oldest-old participants in CREW allows us to add to this current knowledge. We reveal that the oldest-old individuals are at significantly greater risk of poor QoL compared to the youngest-old, for up to five years following their CRC diagnosis. Our findings also demonstrate associations between increasing age and an increased likelihood of reporting problems with mobility and usual activities across five years, which again, is most pertinent for the oldest old. The influence of biological ageing likely underlies these findings. Indeed, the potentially additive effects of cancer and ageing on physical functioning have previously been described [36]. In relation to multiple cancer types, Pergolotti *et al* previously described poorer QoL in older adults reporting lower levels of activities and functioning [35]. These associations may therefore help to explain why the oldest-old experience poorer QoL. Indeed it is likely that this relationship between reduced mobility, daily activities and QoL is likely to reflect the desire of older individuals to maintain their independence and participation in valued recreational activities [37].

With CRC incidence rates highest in the 85–89 year age group, our findings hold importance and relate to the need for development of interventions tailored to the oldest-old [12, 38]. Holistic approaches to personalised cancer care which consider the impact of CRC on mobility and daily activities are key. Supported self-management strategies which are aligned with an individual's wider needs (e.g. mobility) and goals (e.g. independence) may help to address this deficit in QoL for the oldest-old [39].

## Self-efficacy

For older individuals, levels of confidence to manage illness-related problems (self-efficacy) reduce with increasing age, with the oldest-old reporting significantly lower levels of self-efficacy. This is important, because lower self-efficacy demonstrated significant associations with poorer QoL and worse problems across all five health status domains (mobility, self-care, usual activities, pain/discomfort, anxiety/depression) throughout the 5-year follow-up period. In support of our findings, higher self-efficacy has previously been associated with reduced frailty in older adults and better QoL [40]. In addition to the reasons detailed above, lower levels of self-efficacy in the oldest-old may also help to explain why they experience poorer QoL.

Whilst previous research into the self-efficacy of the total CREW cohort demonstrated improvements in self-efficacy over time [29], our findings here suggest that an individual's self-efficacy at the time of diagnosis is of specific importance in determining future health and wellbeing outcomes. The modifiable nature of self-efficacy makes this a particularly pertinent finding, drawing attention to self-efficacy as a potential future target for intervention. Such interventions directed at the point of diagnosis, particularly in the oldest-old, are likely to improve patient experiences, long into CRC recovery.

## Social support

The prevalence of living alone increased with each age sub-group at baseline, being twice as high among the oldest-old compared with the youngest-old. The oldest-old also reported significantly lower levels of social support than the youngest-old, highlighting this as a particular vulnerability for this group [12].

Interestingly, only perceived social support at baseline, and not living alone status, was significantly associated with better QoL, anxiety/depression and usual activities over time. Whilst statistically, social support may have accounted for living alone status, our findings support the call to strengthen the efficacy of healthcare practices in addressing the social support needs of older people living with cancer [12]. We go one step further, and with levels of social support lowest among the oldest old, highlight this as a particularly pertinent issue for this age group. With perceived social support of significance to outcomes, it is important for health and social care professionals to signpost all older people living with CRC, not just those living alone, to support networks and connections.

## Comorbidities

The significance of limiting comorbidities (comorbidities which participants reported 'limited their typical daily activities', most commonly 'arthritis or rheumatism' and 'depression or anxiety') has previously been described in the total CREW cohort in relation to QoL, symptom and functioning outcomes [23]. We add to these findings by revealing the significant burden of limiting comorbidities not just on QoL, but also self-rated health status domains, particularly among older CREW participants. Specifically, the presence of limiting comorbidities was negatively associated with problems with mobility, usual activities, and pain/discomfort over time. Our findings support the assessment of comorbidities as part of personalised care plans for older people with cancer [12, 39]. Moreover, we propose the assessment of whether comorbidities limit someone's typical daily activities, as an important predictor of health and wellbeing outcomes.

## Strengths and limitations of the study

Strengths of our study include its longitudinal design and inclusion of pre-treatment baseline assessments, enabling repeated measures to be assessed before colorectal cancer treatment and across a five-year follow-up period. Further value is in the representativeness of the CREW sample and our inclusion of the oldest old individuals. Whilst loss to follow-up is inevitable in large cohort studies, 72% of older participants still eligible for follow-up, returned questionnaires at five years. The self-report nature of questionnaires excluded non-English speakers from participation.

## Conclusions

This paper reports QoL and self-reported health status in older people living with CRC. Our results indicate the pre-surgery characteristics of individuals which predict poorer outcomes over time: the oldest-old are at risk of poorer health and wellbeing in the five years following CRC surgery, as are those with poor self-efficacy, low levels of social support and limiting comorbidities. Importantly, several of these predictors, self-efficacy, social support and limiting comorbidities, are amenable to intervention and improvement. Further research is recommended to design and evaluate interventions aimed at supporting older people living with CRC whose pre-surgery characteristics predict poorer outcomes. If they are supported and managed, through tailored personalised care planning and appropriate interventions from the time of diagnosis, significant improvements could be made to the quality of survivorship for older adults living with CRC globally. It is especially important to provide health care that is responsive to the needs and experiences of older people living with cancer, as the incidence rises in an ageing population.

## Supporting information

**S1 Appendix. Significant predictors (p<0.05) in the first regression model for each outcome separately by domain.**
(DOCX)

**S2 Appendix. Distribution of baseline characteristics for older (≥65 years) and younger (<65 years) CREW participants.**
(DOCX)

**S3 Appendix. The distribution of QLACS-GSS for older participants at each timepoint in CREW.**
(DOCX)

**S4 Appendix. The distribution of older participants reporting any health problems by five EQ-5D domains at each timepoint in CREW.**
(DOCX)

## Acknowledgments

We thank all CREW study participants and recruiting NHS Trusts; Carol Hill, Kerry Coleman, Bjoern Schukowsky, Christine May (study support); Matthew Breckons, Cassandra Powers, Alex Recio-Saucedo, Bina Nausheen, Ikumi Okamoto, Kim-Chivers Seymour, Joanne Haviland (researchers); Jo Clough, Alison Farmer (research partners). Members of the Study Advisory Committee: Jo Armes, Janis Baird, Andrew Bateman, Nick Beck, Graham Moon, Claire Hulme, Peter Hall, Karen Poole, Susan Restorick-Banks, Paul Roderick, Claire Taylor, Jocelyn Walters, Fran Williams, Lynn Batehup, Jessica Corner, and Deborah Fenlon.

## Author Contributions

**Conceptualization:** Amanda Cummings, Rebecca Foster, Lynn Calman, Natalia V. Permyakova, Jackie Bridges, Theresa Wiseman, Teresa Corbett, Peter W. F. Smith, Claire Foster.

**Data curation:** Lynn Calman, Natalia V. Permyakova, Claire Foster.

**Formal analysis:** Natalia V. Permyakova.

**Funding acquisition:** Lynn Calman, Claire Foster.

**Investigation:** Lynn Calman, Claire Foster.

**Methodology:** Amanda Cummings, Rebecca Foster, Lynn Calman, Natalia V. Permyakova, Jackie Bridges, Theresa Wiseman, Teresa Corbett, Peter W. F. Smith, Claire Foster.

**Project administration:** Lynn Calman, Claire Foster.

**Supervision:** Lynn Calman, Peter W. F. Smith, Claire Foster.

**Visualization:** Amanda Cummings, Natalia V. Permyakova.

**Writing – original draft:** Amanda Cummings, Rebecca Foster, Natalia V. Permyakova.

**Writing – review & editing:** Amanda Cummings, Rebecca Foster, Lynn Calman, Natalia V. Permyakova, Jackie Bridges, Theresa Wiseman, Teresa Corbett, Peter W. F. Smith, Claire Foster.

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
