## [Decision Letter · Decision Letter 0]

7 Sep 2021

PONE-D-21-23944Quality of life and health status in older adults (>65 years) up to five years following colorectal cancer treatment: Findings from the ColoREctal Wellbeing (CREW) cohort studyPLOS ONE

Dear Dr. Cummings,

Thank you for submitting your manuscript to PLOS ONE. After careful consideration, we feel that it has merit but does not fully meet PLOS ONE’s publication criteria as it currently stands. Therefore, we invite you to submit a revised version of the manuscript that addresses the points raised during the review process.

We look forward to receiving your revised manuscript.

Kind regards,

Sinan Kardeş, M.D.

Academic Editor

PLOS ONE

 “The ColoREctal Well‐being (CREW) study is funded by Macmillan Cancer Support grant number 3546834”         

5. Thank you for stating the following in the Competing Interests/Financial Disclosure* (delete as necessary) section:

“I have read the journal's policy and the authors of this manuscript have the following competing interests: Dr Lynn Calman has received an honorarium for teaching from Boehringer Ingelheim”

We note that one or more of the authors are employed by a commercial company: “Boehringer Ingelheim”

“The funder provided support in the form of salaries for authors “L.C” but did not have any additional role in the study design, data collection and analysis, decision to publish, or preparation of the manuscript. The specific roles of these authors are articulated in the ‘author contributions’ section.”

Reviewers' comments:

Reviewer's Responses to Questions

**Comments to the Author**

1. Is the manuscript technically sound, and do the data support the conclusions?

Reviewer #1: Yes

Reviewer #2: Yes

2. Has the statistical analysis been performed appropriately and rigorously? 

Reviewer #1: Yes

Reviewer #2: I Don't Know

3. Have the authors made all data underlying the findings in their manuscript fully available?

Reviewer #1: Yes

Reviewer #2: No

4. Is the manuscript presented in an intelligible fashion and written in standard English?

Reviewer #1: Yes

Reviewer #2: Yes

5. Review Comments to the Author

Reviewer #1: This is a well-designed cohort study. But there is some minor disadvantages.

1.In the Abstract- Objectvie part, the author defined the age of the older participants to be greater than 75 years old, which is inconsistent with the whole manuscript.

2.In line 108, the logo *Insert Figure 1* should be delete.(in line 236, the logo*Insert Table 1* , and in line 277).

3.The statistical part should be carefully checked by a professional statistician.

4.In the Results-Participants part, it would be better to have a detailed screening flowchart.

5.In line 301, the author wrote “Adequate social support was predictive of less problems with usual activities (OR=0.62, CI95% 0.42 to 0.91) and anxiety/depression (OR=0.56 CI95% 0.37 to 0.84)”. The format in the brackets can be modified to (OR=0.62, 95%CI: 0.42- 0.91), and the author should carefully modify other expressions.

6.The author should double check the reference format. For example, the reference 09.

7.For the tables in the manuscript, is it more appropriate to use a three-line table?

8.In the table 1, could you give test statistics, such as t-value or chi-square value?

9.The figure 1 legends could deleted “(19)”. And the Figure 1. should be revised as “Figure 1”. The full name of “LQ” and ” UQ” should be given in the last note.

10.The title of each paragraph could delete the underline. For example, Abstract should revised as Abstract. AND Introduction ...

Reviewer #2: Overall good premise and exploratory study, valuable information when dealing with elderly CRC patients.

lots of data - is it possible to make it more succinct? Suggest making it easier to read for a surgical audience

Suggest bold and definitive conclusive statements to define your point for each section - at the moment is reads as a lot of information but I am not being guided as to what you are specifically trying to say in each paragraph, 'lost in the direction of the narrative' through the results and discussion a little.

Clinical relevance vs statistical relevance of some covariates (i,e, at least one limiting comorbidity) - this is confusing for the reader, i.e. is a limiting comorbidity hypertension vs stroke for example. In results - would be good to see p-values for your OR(CI95%), for the key defining results (i note that its already on table under *, ** etc).

6. PLOS authors have the option to publish the peer review history of their article (what does this mean?). If published, this will include your full peer review and any attached files.

Reviewer #1: No

Reviewer #2: No

---

## [Author Response · Author response to Decision Letter 0]

17 May 2022

Thank you for your helpful comments, please see the Response to Reviewers documents for a detailed response

---

## [Editor Report · Decision Letter 1]

3 Jun 2022

Quality of life and health status in older adults (>65 years) up to five years following colorectal cancer treatment: Findings from the ColoREctal Wellbeing (CREW) cohort study

PONE-D-21-23944R1

Dear Dr. Foster,

We’re pleased to inform you that your manuscript has been judged scientifically suitable for publication and will be formally accepted for publication once it meets all outstanding technical requirements.

Kind regards,

Sinan Kardeş, M.D.

Academic Editor

PLOS ONE

---

## [Editor Report · Acceptance letter]

6 Jul 2022

PONE-D-21-23944R1 

Quality of life and health status in older adults (>65 years) up to five years following colorectal cancer treatment: Findings from the ColoREctal Wellbeing (CREW) cohort study 

Dear Dr. Foster:

I'm pleased to inform you that your manuscript has been deemed suitable for publication in PLOS ONE. Congratulations! Your manuscript is now with our production department. 

Kind regards, 

on behalf of

Dr. Sinan Kardeş 

Academic Editor

PLOS ONE